# MobileNetV2 Combined with Fast Spectral Kurtosis Analysis for Bearing Fault Diagnosis

**Tian Xue, Huaiguang Wang * and Dinghai Wu**

Department of Vehicle and Electrical Engineering, Shijiazhuang Campus of Army Engineering University of PLA, Shijiazhuang 050003, China
* Correspondence: jxxywhg1355@sohu.com

**Abstract:** Bearings are an important component in mechanical equipment, and their health detection and fault diagnosis are of great significance. In order to meet the speed and recognition accuracy requirements of bearing fault diagnosis, this paper uses the lightweight MobileNetV2 network combined with fast spectral kurtosis to diagnose bearing faults. On the basis of the original MobileNetV2 network, a progressive classifier is used to compress the feature information layer by layer with the network structure to achieve high-precision and rapid identification and classification. A cross-local connection structure is added to the network to increase the extracted feature information to improve accuracy. At the same time, the original fault signal of the bearing is a one-dimensional vibration signal, and the signal contains a large number of non-Gaussian noise and accidental shock defects. In order to extract fault features more efficiently, this paper uses the fast spectral kurtosis algorithm to process the signal, extract the center frequency of the original signal, and calculate the spectral kurtosis value. The kurtosis map generated by signal preprocessing is used as the input of the MobileNetV2 network for fault classification. In order to verify the effectiveness and generality of the proposed method, this paper uses the XJTU-SY bearing fault dataset and the CWRU bearing dataset to conduct experiments. Through data preprocessing methods, such as data expansion for different fault types in the original dataset, input data that meet the experimental requirements are generated and fault diagnosis experiments are carried out. At the same time, through the comparison with other typical classification networks, the paper proves that the proposed method has significant advantages in terms of accuracy, model size, training speed, etc., and, finally, proves the effectiveness and generality of the proposed network model in the field of fault diagnosis.

**Keywords:** lightweight network; classifier structure; cross local connection; fast kurtogram; fault diagnosis

## 1. Introduction

Bearings are widely used in mechanical equipment as basic components, and their running state has a significant impact on the normal operation of the entire mechanical system. Therefore, it is very important to carry out fault diagnosis and life prediction for rolling bearings. In recent years, under the background of the vigorous development of big data and artificial intelligence technology, mechanical condition monitoring and fault diagnosis methods based on deep learning have received more and more extensive attention.

The implementation of deep learning algorithms requires the support of high-performance GPUs, which is difficult for meeting the needs of real-time use [1], and it is even more difficult to achieve real-time applications when using embedded devices with weak computing power [2]. In order to reduce the parameters and calculation speed of the network, the field of deep learning has expanded the research on lightweight networks. The earliest lightweight neural network is the Squeeze Net, which was proposed in 2016. Squeeze Net maintains the accuracy of the Alex Net [3] network, and the model parameters are

only 1/540 of that of the Alex Net. After that, a series of lightweight networks such as Shuffle Net [4], Xception [5], and Mobile Net [6] have been developed one after another. While ensuring accuracy, the volume is smaller, and the speed is faster. The advent of these models gives us the opportunity to run deep learning models on portable devices in the future.

In view of the advantages of the lightweight network in terms of parameter quantity and operation speed, a large number of scholars have applied the lightweight network to related work, such as image classification. Liu et al. [7] used MobileNetV2 as the research object and used the model distillation algorithm to optimize the training, and the accuracy rate for the experimental object reached 97.9%. Gao et al. [8] took the lightweight network Eficient Net as the main body, used transfer learning and a new attention mechanism to optimize the model, and achieved an accuracy of 96.17% in Huawei cloud recognition tasks. Meng et al. [9] used the MobileNetV2 network to replace the pyramid convolutional neural network and added an encoder–decoder structure to improve the output image resolution and implemented an improved MobileNetV2 network-based semantic segmentation algorithm. Akhenia et al. [10] considered that when deep learning technology is applied to the field of fault diagnosis, the training of the model requires the support of a large amount of data. To solve this problem, Akhenia used a single-image generative adversarial network (SinGAN) as a data augmentation technique and combined it with the MobileNetV2 network to achieve high classification accuracy detection of bearing fault severity on the CWRU dataset. Guan et al. [11] constructed a model consisting of MobileNetV2 DCNN and two FC neural network layers to complete the fault classification task of 1000 images within 25 s, with an accuracy rate of 98%, which is higher than that of traditional image classification methods.

In the lightweight network, the MobileNet network is a lightweight network which was proposed by Sifre et al. after studying the separability of deep convolutions. The network greatly reduces calculation amount and the network parameters on the premise of ensuring the calculation accuracy. Later, in 2018, Howard et al. improved MobileNet and designed MobileNetV2 [12], which decomposes traditional convolution operations into sparse expressions with less redundant information based on the depthwise convolution separable technique. In the above, the design of linear bottleneck layer and inverted residuals significantly improves the accuracy of the network's prediction.

On the basis of previous research, this paper designs an improved MobileNetV2 network, uses a progressive classifier, adds a cross-local connection structure to improve MobileNetV2 network's ability to identify faults, and is also more in line with the classification and diagnosis of bearing faults. At the same time, considering that the original signal of mechanical fault often contains a lot of noise, the slight shock signal in the bearing vibration will be buried in the background noise. In order to improve the ability of the proposed method to identify fault signals, this paper combines the fast spectral kurtosis algorithm with MobileNetV2. In view of the characteristic of spectral kurtosis being sensitive to transient shock signals, the fault signal in the original mechanical signal is extracted, and the kurtosis map is made and used as the input of the neural network for fault diagnosis.

## 2. MobileNet Network

### 2.1. MobileNet Network Structure

The network structure of MobileNetV2 [12] is shown in Table 1, which consists of convolutional layers, pooling layers, and a series of bottleneck blocks. Among them, Conv2d is a standard two-dimensional convolution operation, Bottleneck is a bottleneck block composed of reverse residual blocks, Avgpool is a global average pooling operation, $t$ is a channel expansion factor (multiple of channel expansion in Bottleneck), $c$ is the number of output channels, $n$ is the number of repeated iterations of Bottleneck, and $s$ is the step size. Replacing the connection layer of 1280 neurons in the above way can effectively prevent overfitting.

**Table 1.** MobileNetV2 network structure.

| Input | Operation | $t$ | $c$ | $n$ | $s$ |
|-------|-----------|-----|-----|-----|-----|
| $224^2 \times 3$ | Conv2d | - | 32 | 1 | 2 |
| $112^2 \times 32$ | Bottleneck | 1 | 16 | 1 | 1 |
| $112^2 \times 16$ | Bottleneck | 6 | 24 | 2 | 2 |
| $56^2 \times 24$ | Bottleneck | 6 | 32 | 3 | 2 |
| $28^2 \times 32$ | Bottleneck | 6 | 64 | 4 | 2 |
| $14^2 \times 64$ | Bottleneck | 6 | 96 | 3 | 1 |
| $14^2 \times 96$ | Bottleneck | 6 | 160 | 3 | 2 |
| $7^2 \times 160$ | Bottleneck | 6 | 320 | 1 | 1 |
| $7^2 \times 320$ | Conv2d $1 \times 1$ | - | 1280 | 1 | 1 |
| $7^2 \times 1280$ | Avgpool $7 \times 7$ | - | - | 1 | - |
| $1 \times 1 \times 1280$ | Conv2d $1 \times 1$ | - | class | - | - |

*2.2. Depthwise Separable Convolution*

A major feature of the MobileNet network is depthwise separable convolution, which can be disassembled into single-channel convolution and point-by-point convolution. In the concept of traditional convolution, the input image of the network needs to be convolved with each convolution kernel, and the number of channels of the output image is also affected by the number of convolution kernels. In the MobileNet network, the use of single-channel convolution makes each convolution kernel responsible for only one channel of the input image, so the resulting output image has the same number of channels as the input image. There is no essential difference between point-by-point convolution and traditional convolution in operation, but a $1 \times 1$ convolution kernel is used. The depthwise convolutional separable structure is shown in Figure 1 below.

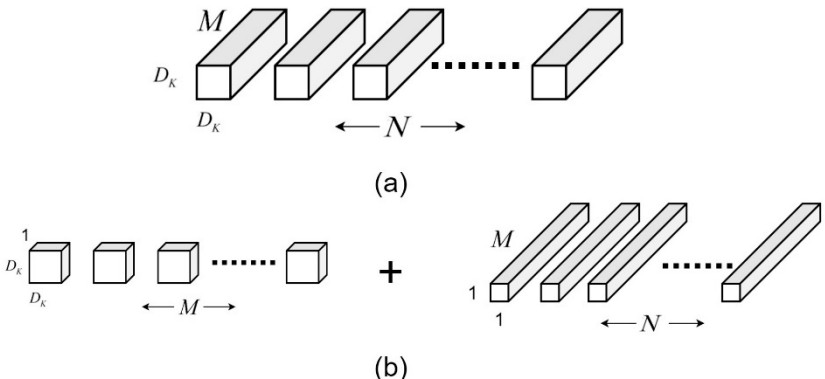

**Figure 1.** Depthwise convolution separable. (**a**): Standard convolutional filters (**b**): Depthwise convolutional filters.

The use of the separable structure of the depthwise convolution makes the MobileNet network have the advantage of saving a lot of computation compared to the traditional convolutional network.

This is assuming that the width and height of the network input image are $D_F$, the number of channels is M, the size of the convolution kernel is $D_K \times D_K$, and the output image has N channels. In traditional convolution, the calculation amount is:

$$C_1 = D_K \times D_K \times M \times N \times D_F \times D_F \tag{1}$$

In the depthwise convolution separable, because single-channel convolution and point-by-point convolution are used, the computation amount is:

$$C_2 = D_K \times D_K \times M \times D_F \times D_F + M \times N \times D_F \times D_F \tag{2}$$

Comparing the two:

$$\frac{C_2}{C_1} = \frac{1}{N} + \frac{1}{D_K{}^2} \tag{3}$$

Taking the $3 \times 3$ convolution kernel commonly used in MobileNetV2 as an example (that is, taking $D_K = 3$), when the output is the same size, and after adopting the structure of single-channel convolution and point-by-point convolution (which are used in depthwise convolution separable), it saves about nine times of computation compared to traditional convolution. Therefore, after adopting the depthwise convolutional separable structure, although the number of network layers increases, the operation speed is significantly improved.

### 2.3. Improve the Network Structure

To better enhance the network's ability to extract fault features and perform fault diagnosis on bearings, this paper uses a progressive classifier and a cross-local connection network backbone structure on the basis of the MobileNetV2 network, and, finally, builds a network model with strong fault diagnosis ability.

1.  Improve the classifier

The classification and recognition capability of the original MobileNetV2 is derived from using the network backbone to extract target features, using the classifier to classify and recognize the output of the last bottleneck. In specific use, the classification and recognition of a specific number of targets can be achieved by modifying the last layer of the classifier between different classification tasks, which is a common, simple, and direct use method. However, when the difference in the number of target classifications between different tasks is too large, the current task goal cannot be achieved just by adjusting the number of neurons in the last layer, and the feature recognition ability of the neural network cannot be fully utilized.

Considering that the original MobileNetV2 network is used to identify more than 1000 types of objects on the ImageNet dataset [13] and that there are six bearing state types involved in this paper, therefore, in order to improve the network's ability to identify fault states, this paper refers to the paper in [14] to redesign the network classifier. The new classifier contains two convolutional layers, one global pooling layer, and one output layer. The specific description is shown in Table 2.

**Table 2.** Improved classifier parameters.

| Input | Operation | Output |
|---|---|---|
| $7 \times 7 \times 320$ | $1 \times 1$ Conv2d, ReLU6 | $7 \times 7 \times 192$ |
| $7 \times 7 \times 192$ | $2 \times 2$ Conv2d, ReLU6 | $6 \times 6 \times 64$ |
| $6 \times 6 \times 64$ | Avgpool | $1 \times 1 \times 64$ |
| $1 \times 1 \times 64$ | Conv2d | $1 \times 1 \times 5$ |

The classifier can convert the features extracted by the network backbone into specific classification results. Because the number of classifications in this paper is far from the dimension of the feature map output by the backbone, two convolution kernels of different sizes are selected to replace the single convolution kernel in the original classifier for feature map compression and conversion.

The structure first has a convolution kernel of size $1 \times 1$, which is responsible for the channel number compression of the feature map. In this layer, 3/5 of the original number of channels is reserved, that is, 192 layers are reserved to prevent the loss of features caused by a large compression rate. The second convolution kernel is used for feature map size compression. In order to avoid subsequent global pooling fluctuations on larger feature maps, this layer compresses the number of channels to 64 layers. The global pooling layer can extract feature information and output the recognition result in the last layer.

2.  Improve the network backbone

The underlying structure of the neural network mostly extracts common features, such as shape, color, etc. Bottleneck uses the residual structure, but if the Bottleneck is used for the entire network backbone, the network will save less basic features due to the deep depth, which will affect the recognition effect. To enhance the fault diagnosis capability of the network for bearings, this paper uses a cross-local-link-network backbone. Based on the original network structure, the design changes the fourth and sixth Bottlenecks, uses convolution and pooling operations to replace the fourth Bottleneck, and adds a cross-local link at the sixth Bottleneck. The improved network consists of two parts: backbone and classifier. The specific structure is shown in Figure 2.

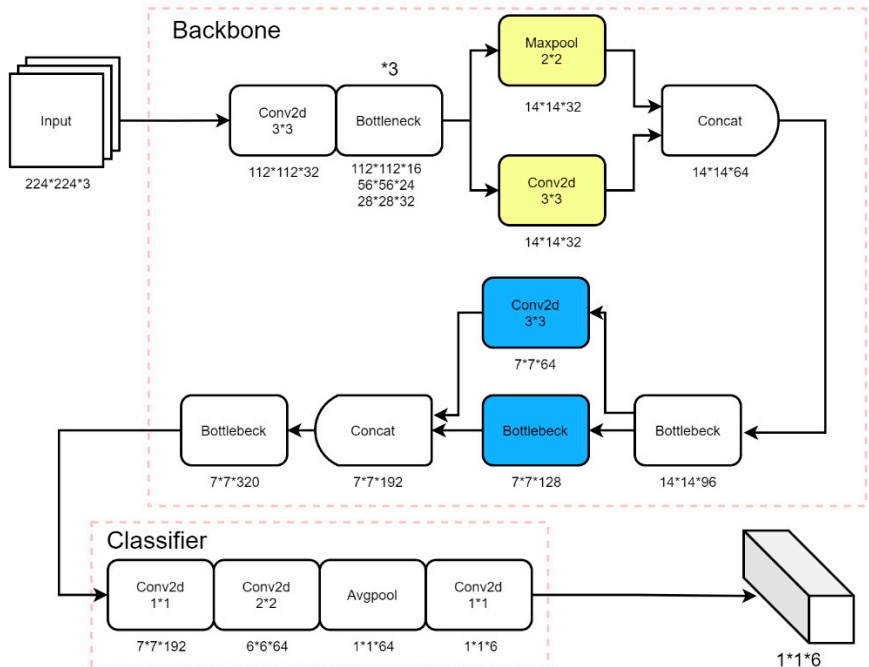

**Figure 2.** Improved MobileNetV2 network structure.

In Figure 2, there is a mark of *3 on the first Bottleneck. This means that there are three Bottleneck operations there. Each operation in the figure has data to mark the current feature map size, and Conv2d boxes refer to convolution kernels. In the figure, the background colors of the squares are marked as yellow and blue, which are the fourth and sixth Bottleneck structures that were replaced, as mentioned above.

After passing through the first three Bottlenecks, convolution and pooling operations are performed on the output of the previous layer. The extracted features of the two are combined and passed to the fourth Bottleneck. At the same time, the number of channels of the fifth Bottleneck are reduced from 160 to 128. The input features are combined with the output through a single-layer convolution to form a local link, so that the input features in the final Bottleneck have different scale information to further enhance the network's ability to identify and classify different targets.

## 3. Fast Spectral Kurtosis Principle

### 3.1. Spectral Kurtosis

In 1983, Dwyer [15] first proposed the concept of spectral kurtosis. Its essence is to calculate the high-order statistics of the kurtosis value of each spectral line. The spectral kurtosis is very sensitive to the transient shock in the signal, and it can effectively identify transient impulses and their distribution in frequency bands from signals containing background noise. At present, considerable progress has been made in the field of fault diagnosis by applying spectral kurtosis [16–19].

After theoretical research on spectral kurtosis, Anyoni gave the definition of spectral kurtosis through Wold-Cramér decomposition and applied it in the field of fault diagnosis.

The expression of the nonstationary signal obtained by Wold–Cramér decomposition given by Anyoni [20] is as follows:

$$y_{(t)} = \int_{-\infty}^{+\infty} e^{j2\pi ft} H(t, f) dx(f) \tag{4}$$

In the above formula: $y_{(t)}$ is the nonstationary signal; $H(t, f)$ is the time-varying transfer function, and it is the complex envelope of $y_{(t)}$ at the $f$ frequency. According to the definition of spectral order moment, the definition of spectral kurtosis is as follows:

$$K_y(f) = \frac{c_{4y}(f)}{s_{2y}^2(f)} = \frac{s_{4y}(f)}{s_{2y}^2(f)} - 2f \neq 0 \tag{5}$$

In the formula: $c_{4y}(f)$ is the fourth-order spectral cumulant of the nonstationary signal $y_{(t)}$ and $s(f)$ is the instantaneous moment of the spectrum.

When reflecting the detection effect of spectral kurtosis, the output result will be expressed as the kurtosis value at the $f$ frequency. The results are as follows:

$$K_{y+\varphi}(f) = \frac{K_y(f)}{[1 + \rho(f)]^2} \tag{6}$$

In the formula: $\varphi(t)$ is the Gaussian signal noise; $\rho(f)$ is the reciprocal of the signal-to-noise ratio.

As can be obtained from Formula (6), when a signal increases the signal-to-noise ratio, $K_{y+\varphi}(f)$ is closer to $K_y(f)$ and, thus, the most ideal filter frequency band can be obtained.

### 3.2. Fast Spectral Kurtosis Map Algorithm

Antoni et al. [21] proposed a fast spectral kurtosis algorithm in 2007, which simplifies the kurtogram operation process and improves the processing efficiency of the algorithm, which can be applied to solve nonstationary signals.

The fast spectral kurtosis algorithm will create a bandwidth-center frequency array, decompose the signal by the frequency band alternating bisection method or trisection method, and construct a tree-like bandpass filter to achieve bandpass filtering of the time domain signal and calculate the envelope signal, calculate the corresponding kurtosis index, assign the index according to the color map, and, finally, draw a grid diagram about the bandwidth and center frequency.

Taking the bisection method as an example, the process of the fast spectral kurtosis map algorithm [22–24] is shown in the following formula:

(1) Mathematically, a multipass filter and a high-pass filter are established, respectively:

$$h_0(n) = h(n)e^{j\pi n/4}(f \in [0, 1/4]) \tag{7}$$

$$h_1(n) = h(n)e^{j\pi n/4}(f \in [1/4, 1/2]) \tag{8}$$

In the above formula: $h(n)$ indicates a low-pass filter with a cutoff frequency of 1/8, and the corresponding filter frequency bands are (0, 1/4) and (1/4, 1/2), respectively. These two high-pass filters and low-pass filters are used to filter the signal, and then twofold down-sampling is performed to obtain the sub-band signal, as shown in Figure 3. In order to achieve no band dislocation factor, the signal is decomposed and filtered layer by layer according to this method, and 2k signals are obtained at the kth layer.

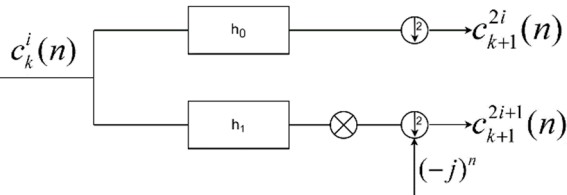

**Figure 3.** Low-pass and high-pass filter decomposition.

(2) Bandpass filter the filtered signal again.

(3) Repeat step (2) several times to acquire a fast spectral kurtosis map.

On the basis of the dichotomy method, the fast spectral kurtosis algorithm of 1/3-binary tree filter is adopted. The first level is shown as a binary tree structure, the second layer will decompose the signal into a 1/3 tree structure, and the remaining layers are obtained by analogy. The overall hierarchical structure description is shown in Figure 4.

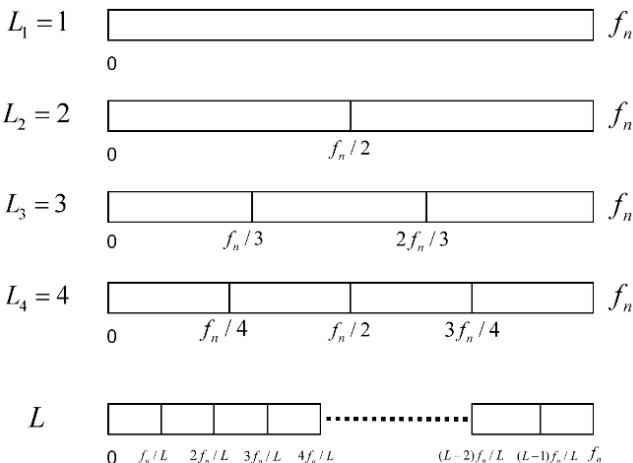

**Figure 4.** Hierarchical Structure of Fast Spectral Kurtosis under 1/3-bindary Tree Filter.

The fast spectral kurtosis method can be applied in the field of fault diagnosis. By this method, the impulse signal can be displayed in the frequency domain, and the estimated values of the center frequency and bandwidth can be obtained.

## 4. MobileNetV2 Combined with Fast Spectral Kurtosis Method

The process of the method proposed in this paper is shown in Figure 5 below, which consists of three parts: data collection, data conversion, and fault classification based on MobileNetV2 network.

As shown in Figure 5, the overall process of the experiment is: obtain different types of bearing fault signals from the bearing simulation signal test bench, obtain a fast spectral kurtosis map for the original signal, analyze its envelope and other information, and extract the kurtosis of the current signal. The value is used as the input of the MobileNetV2 network for classification and fault diagnosis after data processing.

The spectral kurtosis can effectively extract the vibration shock signal from the original signal, and this sensitivity can effectively identify the frequency band of the faulty bearing. In this paper, the spectral kurtosis map of bearing vibration data under various fault conditions is obtained by the fast spectral kurtosis map algorithm and, after processing, the input of the neural network is obtained to train the lightweight network for fault classification.

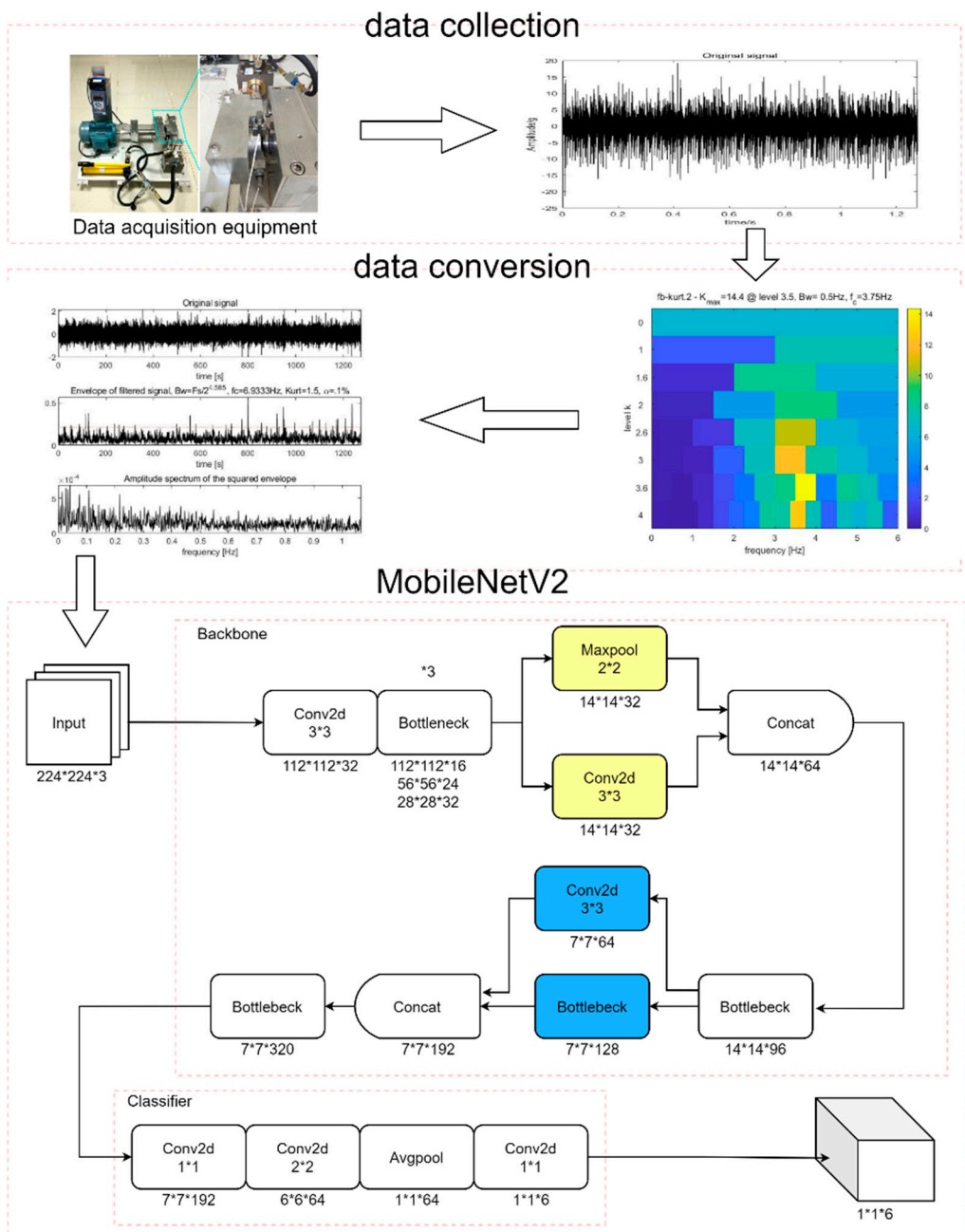

**Figure 5.** Flowchart of the Proposed Method.

*Data Sample Augmentation*

The neural network constructed by deep learning needs to use a considerable amount of data as input in the training process to fully learn the features between different categories. In view of the difficulty of obtaining the original signal in the actual industrial environment, it is practical to expand the signal.

In this paper, the experimental sample expansion method adopted by Zhang Long [25] is used to expand the original data. The method adopts the overlapping sample method to segment and expand the original data, obtains the kurtosis map under single-type fault through data processing, and obtains time series samples through data conversion. The method used for sample augmentation is shown in Figure 6.

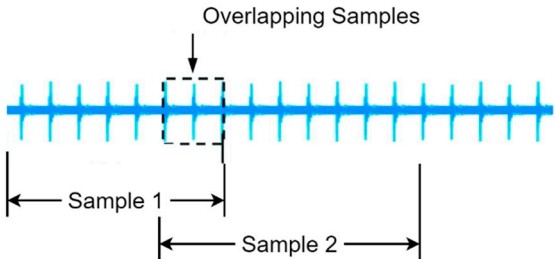

**Figure 6.** Sample Expansion Diagram.

In order to make the signal sample contain as much fault information as possible without being too long, the overlapping sample part needs to select the number of sample points contained in one revolution of the faulty bearing. The specific implementation method is as follows:

$$m = \left[ 60 \times \frac{f_s}{n} \right] \tag{9}$$

$$S = \left[ \frac{N_o - L}{m} + 1 \right] \tag{10}$$

In the above two formulas: $f_s$ represents the fault sampling frequency, the unit is Hz; $n$ is the motor speed, the unit is rpm; $m$ is the sample length of one rotation of the bearing; $N_o$ is the total length of the original signal; $L$ represents the set sample length; and $S$ represents the total number of samples that can be expanded.

## 5. Experimental Data and Comparison

In this paper, two sets of data are used for experimental analysis to verify the effect of the proposed model on the accuracy of fault diagnosis and the generalization of the model. One set of data is the XJTU-SY bearing fault dataset from Xi'an Jiaotong University in China [26], and the other set is the CWRU bearing fault data of Case Western Reserve University.

### 5.1. Dataset 1

There is a total of three working conditions in the XJTU-SY bearing fault dataset. The specific descriptions are shown in Table 3. Each working condition contains five bearing data with different faults, and there are a total of 15 bearing vibration signals covering the whole life cycle in the full dataset.

**Table 3.** Bearing Accelerated Life Test Conditions.

| Condition Number | 1 | 2 | 3 |
|---|---|---|---|
| Rotating speed (r/min) | 2100 | 2250 | 2400 |
| Radial Force /kN | 12 | 11 | 10 |

The data of this dataset are all collected through experiments on the bearing accelerated life test bench. The structure of the test bench is shown in Figure 7 below:

During the data collection process, an LDK UER204 rolling bearing was used as the test bearing, and the vibration signal was collected by the DT9837 portable dynamic signal collector. The sampling frequency in the whole test was 25.6 kHz, the time interval between two samplings was 1 min, and the duration of each sampling was 1 min for 1.28 s. The sampling parameter settings are shown in Figure 8.

Figure 9 shows the bearing pictures of typical failure types, from which it can be seen that the failure causes of the tested bearings include inner ring wear, cage fracture, outer ring wear, outer ring cracks, etc.

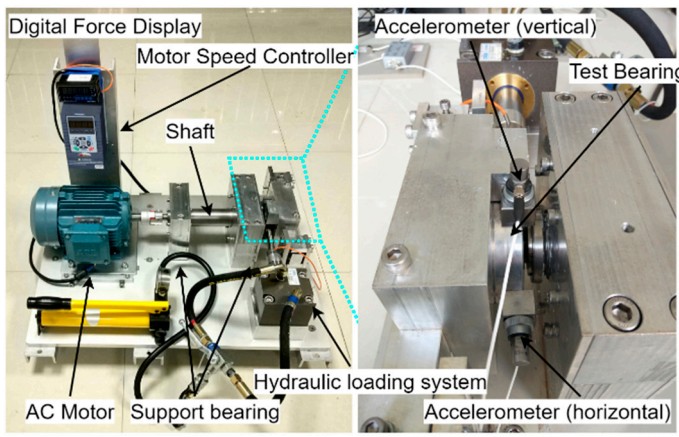

**Figure 7.** Bearing Accelerated Life Test Bench.

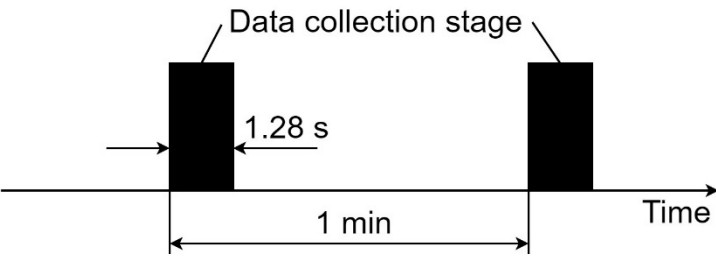

**Figure 8.** Sampling settings for vibration signals.

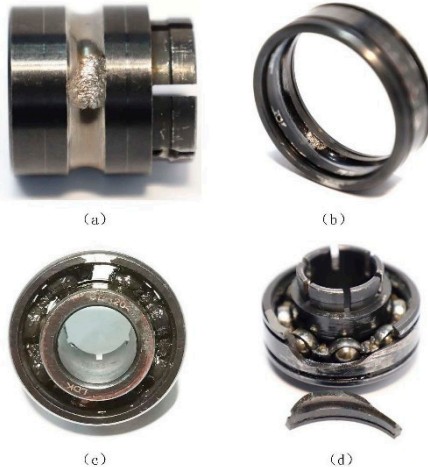

**Figure 9.** Bearing pictures of typical failure types. (**a**): Inner race wear (**b**): Outer race wear (**c**): Cage fracture (**d**): Outer race fracture.

In this paper, two of the three working conditions of the XJTU-SY dataset are taken as experimental data, totaling six categories. Aiming at the problem that the total length of the vibration signal varies among different fault types, the above-mentioned overlapping sampling method is used to process the original data. The specific experimental samples are shown in Table 4. In each fault type, the number of fault samples is 1920, and 1536 of them are randomly selected for use in neural network training, with 384 for testing.

5.1.1. Fast Spectral Kurtosis Algorithm to Find Kurtosis Value

In the experiment, some original signals under working condition 1 are selected, as shown in Figure 10.

**Table 4.** XJTU-SY Dataset Description.

| Bearing Status | Number of Training Sets | Number of Test Sets | Sample Length | Label |
|---|---|---|---|---|
| Condition 1: Outer Ring Failure (1_1) | 1536 | 384 | 1024 | 1 |
| Condition 1: Cage Failure (1_4) | 1536 | 384 | 1024 | 2 |
| Condition 2: Outer Ring Failure (2_2) | 1536 | 384 | 1024 | 3 |
| Condition 2: Cage Failure (2_3) | 1536 | 384 | 1024 | 4 |
| Condition 3: Inner Ring Failure (3_3) | 1536 | 384 | 1024 | 5 |
| Condition 3: Outer Ring Failure (3_5) | 1536 | 384 | 1024 | 6 |

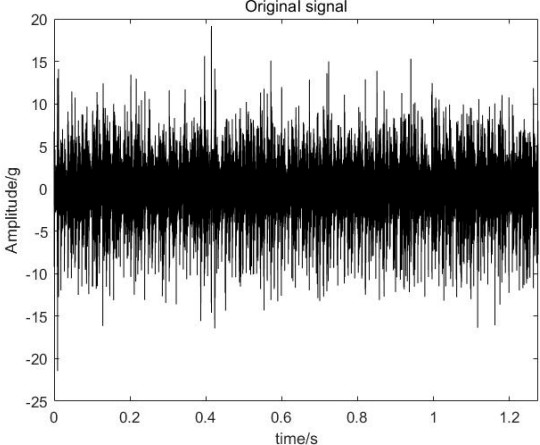

**Figure 10.** Original Signal Map.

The fast kurtosis map is calculated according to the original signal and the resulting image is shown in Figure 11. It can be seen from the image that the maximum spectral kurtosis value of the current sample is 4.3, the level is 3.5, the bandwidth is 10,667 Hz, and the center frequency is 122,667 Hz. From this, the bandwidth and center frequency of the envelope analysis pre filter are determined, and the original signal is passed through the filter for square envelope demodulation analysis.

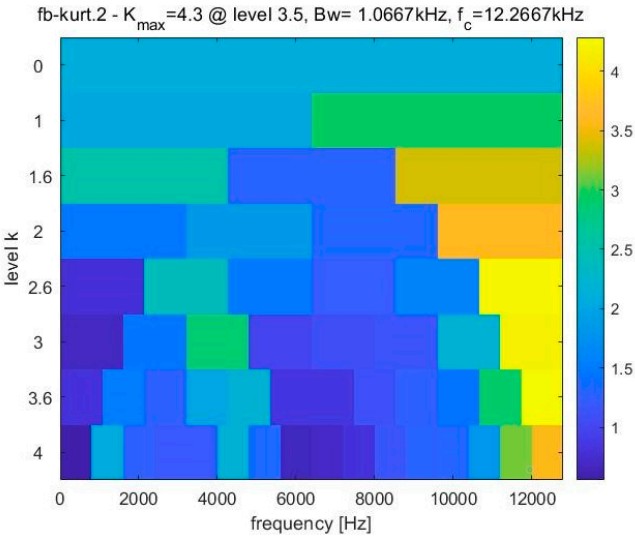

**Figure 11.** Fast Spectral Kurtosis Map.

Figure 12 shows the result of fast spectral kurtosis processing. From top to bottom, the figure shows the original signal, the envelope spectrum calculated by Hilbert transform, and the squared envelope spectrum of the signal obtained after bandpass filtering.

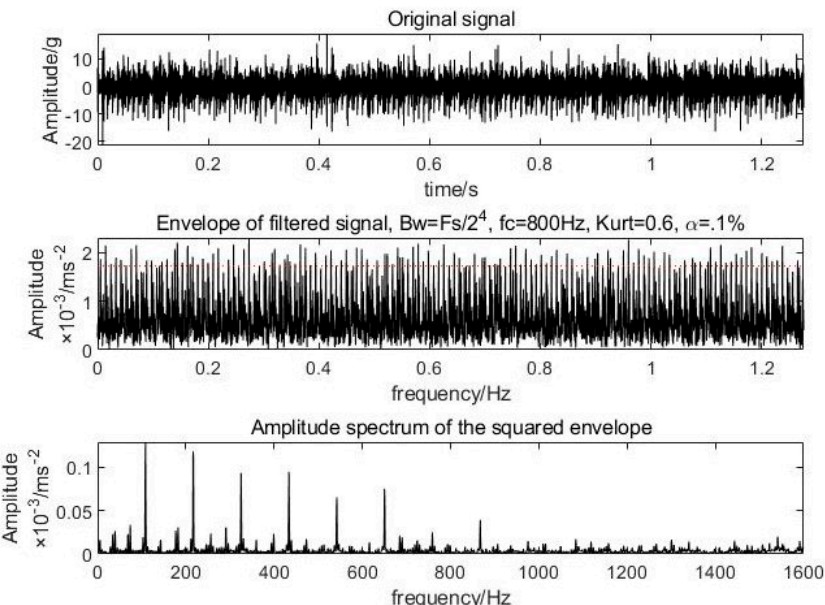

**Figure 12.** Fast Spectral Kurtosis Processing Results.

According to the method described in the neural network input sample above, the processed kurtosis map is used, and the obtained kurtosis map is uniformly cropped to a size of 224 × 224, and, finally, input into the lightweight network.

### 5.1.2. MobileNetV2 for Fault Classification

The six types of experimental samples obtained above are used as input data and the obtained images are input into the MobileNetV2 network. After the data are input, the training set and the test set are divided according to the ratio of 4:1. For the case where each type of fault contains a total of 1920 samples, the number of samples used for training the network is 1536 and the number of samples used for testing is 384.

In order to strengthen the convergence ability of the network, the parameter initialization of the MobileNetV2 network adopts the method of transfer learning, using the weight values pretrained on the ImageNet [13] dataset. The pooling method of the MobileNetV2 network is global average pooling and the optimizer is an adaptive moment estimation optimizer. The learning rate was initialized to 0.001, the cross-entropy function was sampled as the loss function, and the processing volume per batch was 15. The confusion matrix representing the training results is shown in Figure 13.

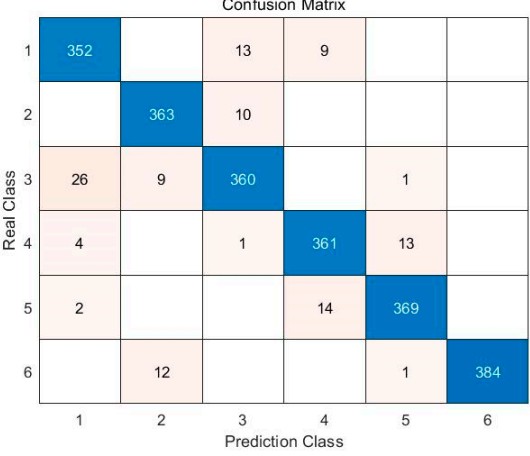

**Figure 13.** Confusion Matrix.

The final model training accuracy is 95%.

### 5.1.3. Optimization and Selection of Hyperparameters

During the training of a neural network model, the generalization performance of the model is affected by the batch size. With the increase of training batches, the training time of the model will decrease and the training curve will be smoother; however, when the batch design is too large, excessive sequence filling will reduce the classification performance of the network. Therefore, a suitable training batch will affect the training effect of the model.

In the text, M is used to represent the training batch, which represents the value of the minibatch. The values of M are 5, 10, 15, and 20, respectively. The accuracy and training loss of the neural network training process under different batches are shown in Table 5. According to the analysis in Table 5, we can get that, when M is selected as 15, the accuracy of the model designed in this paper can reach 95.4% in the case of fewer iterations and training time. Therefore, for the application scenarios designed in this paper, it is more appropriate to select the value M = 15 for the training batch.

**Table 5.** Influence of different batches on training results.

| M | Number of Iterations | Accuracy /% | Loss Rate | Training Time/min |
|---|---|---|---|---|
| 5 | 3680 | 91.4 | 0.1133 | 368 min 40 s |
| 10 | 1840 | 92.8 | 0.2686 | 316 min 36 s |
| 15 | 1220 | 95.4 | 0.0310 | 289 min 5 s |
| 20 | 920 | 93.6 | 0.0964 | 257 min 30 s |

### 5.2. Dataset 2

On the basis of the previous article, this paper uses another set of data to conduct experiments to verify the generality of the proposed method. In the test, to test the universality, the CWRU public bearing fault dataset of Case Western Reserve University is used. The fault test bench obtained from the data is shown in Figure 14.

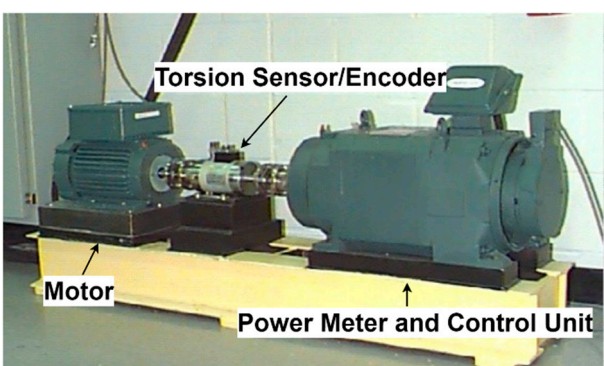

**Figure 14.** CWRU Bearing Failure Test Bench.

As shown, the components included in the test bench are the motor, torsional sensor/encoder, control unit, and power meter. Bearing failure made by EDM.

Similar to dataset 1, six different fault states in the CWUR dataset are selected. The specific fault states of the bearing are shown in Table 6.

Taking the same method as the XJTU-SY dataset, the overlapping sampling method is used to process the six types of fault state data selected from the CWRU dataset, and the experimental samples are finally obtained as shown in Table 7. The number of fault samples of each type is 1920, of which 1536 384 for neural network training and 384 for testing. The 'Inner Ring Failure 7in' in the first column of the table indicates that in this case the fault location is the inner ring, the fault diameter is 7 inches, and so on, in other cases.

**Table 6.** Bearing fault description in CWRU dataset.

| Bearing | Fault Location | Diameter/ Inches | Depth/ Inches | Bearing Manufacturer | Label |
|---|---|---|---|---|---|
| Drive End | Inner Raceway | 0.007 | 0.011 | SKF | 1 |
| Drive End | Inner Raceway | 0.014 | 0.011 | SKF | 2 |
| Drive End | Ball | 0.007 | 0.011 | SKF | 3 |
| Drive End | Ball | 0.014 | 0.011 | SKF | 4 |
| Drive End | Outer Raceway | 0.007 | 0.011 | SKF | 5 |
| Drive End | Outer Raceway | 0.014 | 0.011 | SKF | 6 |

**Table 7.** Description of the samples obtained from the CWRU dataset.

| Bearing Status | Number of Training Sets | Number of Test Sets | Sample Length | Label |
|---|---|---|---|---|
| Inner Ring Failure 7in | 236 | 60 | 1024 | 1 |
| Inner Ring Failure 14in | 236 | 60 | 1024 | 2 |
| Rolling Element Failure 7in | 236 | 60 | 1024 | 3 |
| Rolling Element Failure 14in | 236 | 60 | 1024 | 4 |
| Outer Ring Failure 7in | 236 | 60 | 1024 | 5 |
| Outer Ring Failure 14in | 236 | 60 | 1024 | 6 |

The CWRU dataset is processed using the method described above and the diagnostic results are represented by the confusion matrix of the test set. The fault diagnosis results are shown in Figure 15.

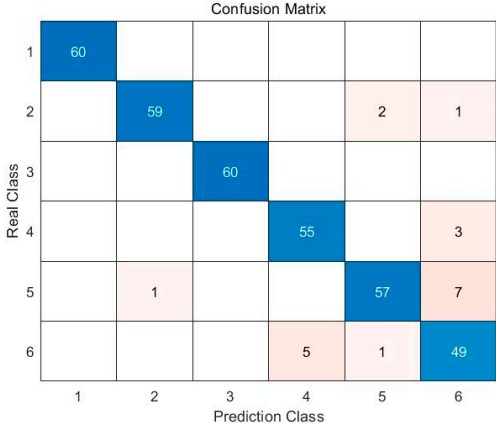

**Figure 15.** Confusion Matrix.

The final model accuracy is 97.8%.

Model Comparison

To verify the effectiveness of the proposed method, three typical classification networks, Xception [5], Resnet-50 [27], and DenseNet-121 [28], were selected for comparison with the method in this paper. Among them, the Xception network uses the Inception module to extract the multiscale features of the target, the Resnet-50 network uses the residual unit structure to solve the common problem of gradient disappearance or explosion in deep learning, and the DenseNet-121 improves the propagation efficiency and utilization efficiency of neural network for feature information. The above three networks have excellent performance in the field of image recognition, so this paper selects the above methods for comparison, which can better reflect the recognition and diagnosis capabilities of this method. When showing the performance comparison between the models, the comparison between the methods is shown in the form of graphs, as follows.

Through analysis in Table 8, it can be seen that the method in this paper is better than Resnet-50 and DenseNet-121 in accuracy, and slightly lower than Xception. In terms of training time, the method in this paper takes significantly less time than other methods. At the same time, compared with Xception, Resnet-50, and DenseNet-121, in terms of network parameters and model size, the parameters of the model proposed in this paper are 8.56%, 25.37%, and 7.57%, respectively, which achieved a great advantage. In terms of recognition speed, the method in this paper is more than three times faster than Xception, Resnet-50, and DenseNet-121.

**Table 8.** Comparison between Different Methods.

| Network | Accuracy /% | Training Time /min | Parameter Quantity | Size/MB |
| --- | --- | --- | --- | --- |
| The Method of This Paper | 97.8 | 58.1 | 1,786,597 | 20.8 |
| Xception | 98.3 | 215.5 | 20,871,725 | 239 |
| Resnet-50 | 92.4 | 168.7 | 7,042,629 | 81.9 |
| DenseNet-121 | 90.8 | 159.2 | 23,597,957 | 270 |

Figure 16 shows the average accuracy and average time-consuming statistics of each model in the test set of the experiment 10 times. In the figure, A is the method in this paper, B is the Xception experiment result, C is Resnet-50, and D is DenseNet-121. From the figure, the accuracy and recognition time of each model can be seen intuitively. It can be seen from the above experiments and analysis that the method proposed in this paper has excellent effectiveness and generality in the field of fault identification.

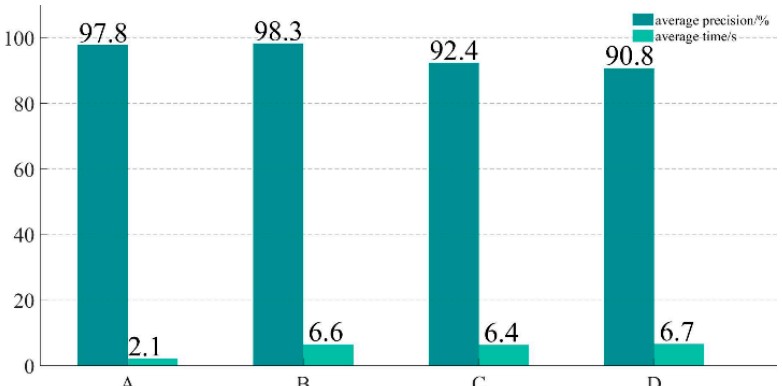

**Figure 16.** Comparing results between different models. A: The method proposed in this paper. B: Xception. C: Resnet-50. D: DenseNet-121.

To sum up, compared with the three networks used, the recognition accuracy of this method is slightly lower than that of Xception, but it has great advantages in training time, parameter quantity, model size, and recognition speed. Among them, the accuracy of Xception is only 0.5% higher than that of the method in this paper, but its recognition speed is not as fast as 1/3 of the method in this paper, and the number of parameters is more than ten times that of the method in this paper. It can be seen that, under the condition of comprehensive consideration of recognition speed and accuracy, the method in this paper has excellent performance. The shorter training time also makes the method in this paper more in line with the needs of practical industrial applications.

In order to better verify the ability of the improved MobileNetV2 network model combined with the fast spectral kurtosis algorithm proposed in this paper to extract fault features, on the basis of the previous experiments, background noise is added to the experimental dataset to verify the application effect of the proposed model in the noise background.

By adding background noise to the six fault signals in the CWRU dataset used in this paper and setting the signal-to-noise ratios (SNR) to −2 dB, −4 dB, −6 dB, −8 dB, and −10 dB, respectively, a dataset containing five different signal-to-noise ratios can be obtained.

In the experiment, the dataset with background noise interference was used as the model input. The experiment was repeated, and the average accuracy of the model obtained in ten experiments was calculated. The Table 9 below shows the accuracy of ten repeated experiments when the signal-to-noise ratio is −10 dB. It can be seen from the table that the improved MobileNetV2 network model combined with fast spectral kurtosis proposed in this chapter has high accuracy and is relatively less affected by noise.

**Table 9.** Comparison between Different Methods.

| SNR/(db) | Xception | Resnet-50 | DenseNet-121 | The Method of This Paper |
|----------|----------|-----------|--------------|--------------------------|
| −2 | 97.9% | 92.5% | 90.1% | 97.5% |
| −4 | 96.2% | 90.6% | 88.3% | 97.6% |
| −6 | 95.7% | 89.4% | 85.3% | 96.9% |
| −8 | 93.2% | 85.2% | 83.9% | 96.2% |
| −10 | 91.8% | 83.4% | 81.9% | 95.9% |

## 6. Conclusions

In this paper, a bearing fault diagnosis method based on the MobileNetV2 network and fast spectral kurtosis technique is proposed. It improves the original MobileNetV2 network structure and uses a progressive classifier to fully learn fault features and convert them into classification results, which improves the recognition accuracy. At the same time, it speeds up the recognition speed and combines the fast spectral kurtosis algorithm to process the spectral kurtosis map to obtain an input signal that conforms to the network format and, finally, inputs it to the MobileNetV2 network for fault diagnosis.

In order to verify the fault diagnosis ability of the proposed network model, the model proposed in this paper is compared with three typical classification network models, namely Xception, Resnet-50, and DenseNet-121. The experiments show that the model proposed in this paper has excellent performance in the field of fault diagnosis and has significant advantages in model training time, network parameters, and model size. At the same time, through the experimental analysis on two sets of datasets, it is proved that the proposed method has high accuracy and generalization.

**Author Contributions:** Methodology, H.W.; validation, D.W.; writing original draft preparation, T.X.; writing—review and editing, H.W., D.W. and T.X. All authors have read and agreed to the published version of the manuscript.

**Funding:** This research received no external funding.

**Data Availability Statement:** The data used to support the findings of this study are available from the corresponding author upon request.

**Conflicts of Interest:** The authors declare that they have no conflicts of interest.

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
