# Peer review of "MobileNetV2 Combined with Fast Spectral Kurtosis Analysis for Bearing Fault Diagnosis"

_electronics, doi:10.3390/electronics11193176_

Round 1

Reviewer 1 Report

this paper uses the fast spectral kurtosis algorithm to process the original signal, extract the center frequency of the signal and calculate the spectral kurtosis value, the kurtosis map generated by signal preprocessing is used as the input of the MobileNetV2 network for fault classification. The effectiveness of the proposed method is verified by comparing with other classification networks.

1. Add more comparative analysis with existing work

2. How the authors perform labeling in the CWRU Dataset 

3. Required the mathematical analysis of all the proposed work 

4. Discuss more about the equation 7 & 8 

5. Rectify all the grammar and typos error  

Reviewer 2 Report

The paper describes research where MobileNetV2 combined with spectral kurtosis processing method deal with fault bearing monitoring. However, there are few things to improve.

Minor things:

1.       The author names of few positions are written by big letters (e.g. [3], [5], etc.) – it should be standardized.

2.       There is a lack of information (e.g. name of the journal, year, etc.) about the [13] reference.

3.       There are gaps between 462-467 verses.

4.       Fig. 1, 3 have a poor resolution – it should be improved.

5.       The text division between 99 and 104 verse looks unprofessional.

6.       Fig. 2 is too small in the normal size of the pdf site; the test is hard to read.

7.       I guess, it should be “Wold-Cramér decomposition” in the 193 and 195 verse. Furthermore, the text in this line is shifted.

8.       The 4-6 equations should be cited as another work, because I think the authors do not invented or modified it.

9.       Please explain what means “the sampling interval is 1mim”?

10.   What kind of accelerometers authors used during the experiment? There is lack of the main technical data description and the measurement chain presentation.

11.   Fig. 7 – Is the vertical y axis description of the acceleration amplitude? It is described in m/s2 or g unit? It should be explained.

12.   Fig. 8 – Could you more precisely describe k level which is equal 3.5 in the maximum spectral kurtosis value?

13.   Fig. 9 – There is lack of vertical axis descriptions on each function (original, filtered, envelope spectrum).

14.   336-337 and 397-402 verses – text shifted. There are also some small editorial mistakes in the text, such as double comma, spaces, etc.

Major things:

Reference statistics: Journals – 15 positions (and only 4 of the references are from high cited journals (e.g. Mechanical Systems and Signal Processing or Artificial Intelligence). Conference proceedings – 11 p. In my opinion authors could show some more other references which are deal with their topic (e.g. usage the MobileNetV2 in other studies or especially few more studies where bearing diagnostics were made also in real conditions not only in laboratory with constant experiment conditions), especially references which are written in the high cited journals (maybe more reliable). It is connected to the Introduction chapter which should be improved.

There is lack of fault description on each bearing (in data set 1 and 2), how big they are? It is important because it is not a problem to find a bearing fault using other (even much simpler) processing methods when it generates a significant vibroacoustic signal. It could be really interesting if proposed method would monitor the technical condition in early stage of damage/wear (where the vibration is very low in defined frequency bands).

The Conclusion chapter is too general, there is lack of the main results short description. Authors should show the main differences between their results and other studies (in case of the experiment methodology and data processing method). Please explain if the proposed method is diagnosing or only monitoring the fault of bearing, in other words the calculation result will show only the damage or will show the type of the bearing damage?

It should be more emphasize the advantages of the research because the bearing fault monitoring based on vibration signal is well-known. In my opinion the weak side of the research is the experimental measurements part, while the processing method and machine learning is well described compared to other solutions.

Reviewer 3 Report

The paper investigates the use of lightweight MobileNetV2 network combined with fast spectral kurtosis to diagnose bearing faults. A progressive classifier is used to compress the feature information layer by layer with the network structure to achieve high-precision and rapid identification and classification, and a cross-local connection structure is added to the network to increase the extracted feature information to improve accuracy. The effectiveness of the proposed method is verified by comparing with other classification networks.

From the point of view of methodology, the paper is well organized. Introduction and methodology parts are well presented and clearly explained. The results are well organized and clear, with some minor corrections needed. The language of the paper is also satisfactory

There are two main issues in the paper I would like to address to.

First of all, the exact novelty of the paper must be emphasized. From the text, one can conclude that only minor modifications of the existing methods (MobileNetV2 and fast spectral kurtosis) are proposed, which is not enough to consider the paper for publication as a research article. It is necessary to strongly emphasize the contribution in the abstract and in the methodology section(s).

Second issue is related to the classification of the data set used for training and testing. You have six classes, containing different faults and different working conditions. In my opinion, such classification is not appropriate. In fault detection, it is crucial to detect and diagnose the fault, regardless of working conditions. Classification needs to address only different faults. It is not the role of the classifier to determine the working conditions (speed, load and so on), but to detect if there is a fault present in any working conditions. In that manner, I suggest to modify the classification (the number and types of output classes) and to output only different faults (outer race, cage and inner race faults). I do not see the significance of determining in which working condition is the fault present, but to detect fault reliably and independently from working condition. It is important to note that the input data set does not need to be changed, it can be used for training and testing in different conditions. In addition, when testing the trained network used to verify the results, you use the other dataset (CWRU set). Is this set obtained for the same working conditions, since you also have 6 classes? Will every new dataset depend on these conditions, or it is possible to use vibration data in different condition? Is you solution then universal, or strictly valid for given conditions? This is something you must have in mind.

There are also some minor issues in the paper I would like to point to:

-          Line 87, parameter k is mentioned, but there is no such parameter in Table 1

-          In Figure 1, you have parts a) and b), but the rest of the figure is neither noted nor explained

-          Figure 2 is not appropriately explained in the text, some parts are omitted

-          Figure 4 is not well commented, input data representation is not fully elaborated

-          When explaining some equations (e.g. Eq (10)), you do not use units for parameters

-          Lines 288-289, the terminology is not correct. What does sampling time refer to? Usually, it is the time between samples, which is connected to sampling frequency, but not in your paper? Please define and explain all parameters

-          In Table 4, what is the meaning of numbers in brackets (1_1, 1_4 and so on)?

-          Figure 7 – there are no units in axes

-          Figure 9 – Units in y-axis are missing

-          Lines 384-385 – what exactly these numbers in percent represent?

-          Table 6 – What is “in” in the first column? If inches, what exactly does this mean? Please explain

-          In general, the font used in figures is very small, it must be increased, along with the quality of some figures.

Round 2

Reviewer 1 Report

In this paper, two sets of data are used for experimental analysis to verify the effect of the proposed model on the accuracy of fault diagnosis and the generalization of the model. add more comparative analysis with existing work. Add the mathematical model  

Reviewer 2 Report

the Tab. 2 is doubled (160-166 verse)

324-325 verses: 1mim or 1 min??

211-213 verses: Wold Carmer or Wold-Cramér? 

274 verse is the same as 293, text shifted.

Reviewer 3 Report

The authors have addressed al the issues and corrected the manuscript according to my suggestions.
